# A single mutation in Crimean-Congo hemorrhagic fever virus discovered in ticks impairs infectivity in human cells

Brian L Hua[1], Florine EM Scholte[1], Valerie Ohlendorf[2,3], Anne Kopp[2,3], Marco Marklewitz[2,3], Christian Drosten[2,3], Stuart T Nichol[1], Christina Spiropoulou[1†], Sandra Junglen[2,3†*], Éric Bergeron[1†*]

[1]Centers for Disease Control and Prevention, Atlanta, United States; [2]Institute of Virology, Charité-Universitätsmedizin Berlin, corporate member of Free University Berlin, Humboldt-University Berlin, and Berlin Institute of Health, Berlin, Germany; [3]German Center for Infection Research (DZIF), Berlin, Germany

**Abstract** Crimean-Congo hemorrhagic fever (CCHF) is the most widely distributed tick-borne viral infection in the world. Strikingly, reported mortality rates for CCHF are extremely variable, ranging from 5% to 80% (Whitehouse, 2004). CCHF virus (CCHFV, *Nairoviridae*) exhibits extensive genomic sequence diversity across strains (Deyde et al., 2006; Sherifi et al., 2014). It is currently unknown if genomic diversity is a factor contributing to variation in its pathogenicity. We obtained complete genome sequences of CCHFV directly from the tick reservoir. These new strains belong to a solitary lineage named Europe 2 that is circumstantially reputed to be less pathogenic than the epidemic strains from Europe 1 lineage. We identified a single tick-specific amino acid variant in the viral glycoprotein region that dramatically reduces its fusion activity in human cells, providing evidence that a glycoprotein precursor variant, present in ticks, has severely impaired function in human cells.

**\*For correspondence:**
sandra.junglen@charite.de (SJ);
ebergeron@cdc.gov (EB)

†These authors contributed equally to this work

**Competing interests:** The authors declare that no competing interests exist.

## Introduction

Crimean-Congo hemorrhagic fever (CCHF) is severe human disease present in an increasing number of regions of Europe, Africa, and Asia (*Bente et al., 2013*). In Turkey alone, more than one thousand endemic CCHF cases are reported annually. Outbreaks of CCHF are sporadic, and reported mortality rates are extremely variable (5–80%) (*Whitehouse, 2004*). The CCHF etiological agent, CCHF virus (CCHFV), is the most widespread tick-borne virus of medical importance and is primarily maintained in and transmitted by hard ticks of the *Hyalomma* genus (*Gargili et al., 2017*). Human infections occur through tick bites or exposure to blood or other bodily fluids of infected animals or CCHF patients.

CCHFV is classified in the family *Nairoviridae* in the genus *Orthonairoviridae*. CCHFV is a negative-sense, single-stranded RNA virus with a genome consisting of three segments called large (L), medium (M), and small (S), which encode for the L RNA-dependent RNA polymerase, the glycoprotein precursor (GPC), and the nucleoprotein (NP), respectively. The GPC is post-translationally processed into several non-structural proteins and the structural glycoproteins, Gn and Gc, that mediate attachment of the virion to the host cell and fusion of the virion and host cell membranes (*Zivcec et al., 2016*). To date, the cellular receptor(s) that mediate CCHFV entry are unknown. While a functional interaction between cell surface nucleolin and CCHFV Gc glycoprotein has been suggested (*Xiao et al., 2011*), further studies are required to test the role of this interaction in the context of CCHFV cellular entry and infection.

**eLife digest** Crimean-Congo hemorrhagic fever (CCHF) is caused by infection with a virus spread by ticks in Europe, Africa and Asia. It can cause severe disease in humans, including high fevers and bleeding. How deadly CCHF is varies with between 5% to 80% of those infected dying. Scientists suspect genetic differences in various strains of the virus may account for the differences in death rates, but they do not know the exact mutations that make the CCHF virus more or less deadly.

To learn more, scientists have sorted strains of CCHF virus into different groups based on how similar they are genetically. One group called Europe 2 infects many people in the Balkans, but it rarely causes illness. In fact, only two mild cases of illness have been associated with Europe 2 strains, while other CCHF virus strains circulating in this region have caused thousands of more severe illnesses.

Now, Hua et al. identified a mutation in one Europe 2 strain of the CCHF virus that may explain why this subgroup of viruses rarely causes severe human disease. The researchers collected a strain of CCHF virus from infected ticks found in Bulgaria and sequenced its genome. They named the virus strain Malko Tarnovo. Through a series of experiments, Hua et al. showed that the Malko Tarnovo strain very efficiently infects tick cells but not human cells. A single amino acid change in the genetic sequence of the virus appears to make the virus less able to infect human cells. The mutation prevents a protein on the surface of the virus from fusing with human cells, an essential step in infection.

This may explain why this strain and others in the Europe 2 group do not cause severe human disease. Hua et al. also demonstrate the importance of studying viruses in the animals that spread them. By studying the CCHF virus in ticks, scientists may be able to learn more about how viruses evolve to infect new species, which may help scientists prevent future pandemics.

---

CCHFV strains exhibit great diversity at the RNA and protein sequence levels, and are divided into seven genetic lineages (Africa 1, 2, and 3; Asia 1 and 2; Europe 1 and 2). All lineages except Europe 2 are believed to be transmitted by *Hyalomma spp.* ticks and cause severe disease in humans. In contrast, the Europe 2 lineage is a phylogenetic outlier that is not usually associated with severe disease. The prototype strain of Europe 2, AP92, was isolated from *Rhipicephalus bursa* ticks collected in Greece in 1975 (*Deyde et al., 2006*; *Papadopoulos and Koptopoulos, 1980*). Since then, Europe 2 strains have only been associated with three documented human cases, including one fatality in Iran and two mild cases in Turkey (*Midilli et al., 2009*; *Salehi-Vaziri et al., 2016*; *Elevli et al., 2010*). Although CCHFV seroprevalence is relatively high in both humans and livestock in the Balkans (*Papa et al., 2011*; *Sidira et al., 2012*; *Sargianou et al., 2013*; *Papa et al., 2014*; *Papa et al., 2016*), almost all clinical cases are caused by Europe 1 strains despite the circulation of Europe 2 strains in ticks and livestock. Thus, given the high CCHFV seroprevalence in areas of Europe 2 circulation and the low number of disease cases associated with these strains, the Europe 2 lineage may contain CCHFV strains that are less pathogenic in humans than strains of other lineages (*Papa et al., 2011*; *Sidira et al., 2012*; *Sargianou et al., 2013*; *Papa et al., 2016*; *Papa et al., 2013*; *Sidira et al., 2013*).

The genomic and molecular properties of CCHFV that directly contribute to its transmission from the tick vector to the human host are largely unknown, partly because only one complete CCHFV genome sequence has been derived directly from a tick (*Cajimat et al., 2017*). All other reported CCHFV genomes are derived from human patients. Thus, studies of the changes in CCHFV genomic signatures upon transmission from ticks to humans or other susceptible species have been precluded. While Europe 2 CCHFV strains have often been detected directly in ticks collected in the Balkans and in Turkey (*Sherifi et al., 2014*; *Papa et al., 2014*; *Panayotova et al., 2016*; *Papa et al., 2017*; *Dinçer et al., 2017*), no full-genome of a Europe 2 strain has been derived directly from a tick to date. Only two Europe 2 CCHFV full-genome sequences have been described, the AP92 and Pentalofos strains, both detected in Greece (*Deyde et al., 2006*; *Papa et al., 2018*), but they were determined after passaging the virus in vertebrate culture systems. Thus, we sought to better address viral gene functions from tick-derived Europe 2 lineage CCHFV in human cells and in the

context of virus transmission from ticks to humans by obtaining a Europe 2 CCHFV sequence directly from ticks.

## Results

### Full-genome sequencing of a Europe 2 CCHFV strain directly from ticks

To obtain a CCHFV sequence directly from ticks, ticks were collected in Strandja Nature Park, a unique pristine forest in Bulgaria. A total of 1541 ticks were collected from vegetation, humans, tortoises, and livestock. CCHFV was detected in three *R. bursa* ticks feeding on one cow from Malko Tarnovo village; the cow showed no obvious signs of disease. Similar CCHFV genome quantities ($1 \times 10^5$ to $1 \times 10^6$ genome copies/ml) were detected in all CCHFV-positive ticks, indicating that the three CCHFV strains replicated similarly in ticks.

Complete CCHFV genomic sequencing directly from tick homogenates and phylogenetic analyses revealed that the CCHFV strains obtained from these three ticks, tentatively named Malko Tarnovo-BG2012-T1302 (MT-1302), Malko Tarnovo-BG2012-T1303 (MT-1303), and Malko Tarnovo-BG2012-T1362 (MT-1362), grouped with other strains of the Europe 2 lineage (*Figure 1A*, *Figure 1—figure supplement 1*). These Malko Tarnovo strains establish a novel clade and share a common ancestor with strains recently detected in Bulgaria and Greece.

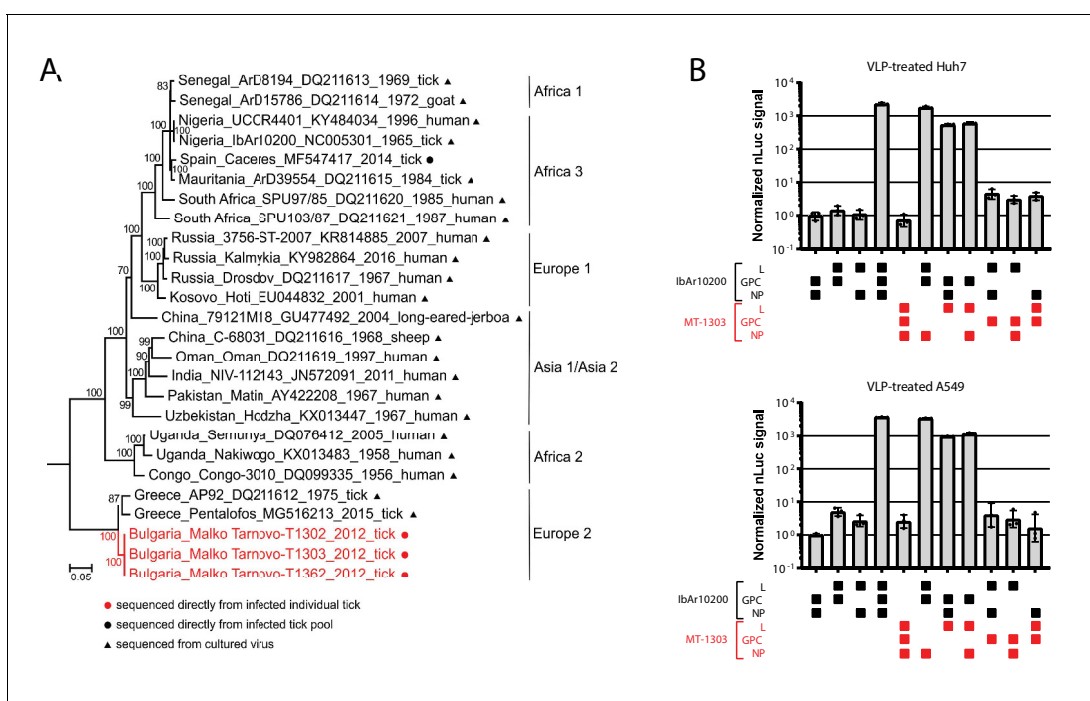

**Figure 1.** Identification of a tick-derived Europe 2 CCHFV genotype that poorly infects human cells. (A) Phylogenetic relationship of Malko Tarnovo strains to other CCHFV strains. Maximum likelihood phylogeny was inferred for the complete coding region of the L-segment. RAxML tree was performed with the GTR substitution model and 1000 bootstrap replicates. Only bootstrap values > 50 are shown. The tree was rooted to Dugbe virus. For each sequence, country, accession number, year of sampling, and host are shown. Sequences derived from virus growing in cell culture are marked by black triangles; sequences derived directly from infected ticks are marked by red circles. Novel sequences are marked in red. (B) Nanoluciferase (nLuc) reporter activity in Huh7 or A549 cells treated with VLPs generated using indicated combinations of viral protein components (L, GPC, and NP) from the IbAr10200 and MT-1303 strains. Error bars represent standard deviation of the mean of two independent biological replicates.

The online version of this article includes the following figure supplement(s) for figure 1:

**Figure supplement 1.** Phylogenetic relationship of Malko Tarnovo strains to other CCHFV strains.

**Figure supplement 2.** CCHFV Malko Tarnovo strains cannot be maintained through serial passaging in virus isolation attempts.

**Figure supplement 3.** MT-1303 L drives replication of minigenomes derived from divergent CCHFV strains.

**Figure supplement 4.** Attenuation of VLP activity is attributed to MT-1303 GPC.

**Figure supplement 5.** Expression and processing of MT-1303 GPC.

Compared to the AP92 and Pentalofos strains, the Malko Tarnovo S segment nucleotide sequences showed 91.9% and 96.5%, the M segment, 86.9% and 96.3%, and the L segment, 96.2% and 96.2% identity, respectively. A detailed comparison between genomes of the Malko Tarnovo strains and prototypes of all lineages, as well as amino acid changes between AP92 and Pentalofos strains, are listed in *Supplementary files 1* and *2*. The Malko Tarnovo sequences are the first full-genome sequences of Europe 2 CCHFV strains obtained from a tick without passaging the virus in a vertebrate culture system (*Supplementary file 3*). In order to analyze the phenotypic properties of the newly identified Malko Tarnovo strains, we attempted virus isolation experiments in different tick and vertebrate cell lines. Unfortunately, we failed to isolate any of the three strains as virus could not be maintained through serial passaging in any of the tested cell lines (*Figure 1—figure supplement 2*).

## The MT-1303 GPC fails to support viral replication in human cells

Next, we examined the ability of Malko Tarnovo strain proteins to support viral replication in human cells. We generated constructs expressing MT-1303 nucleoprotein (NP), glycoprotein precursor (GPC), and L protein, and tested these in a transcription- and entry-competent virus-like particle (VLP) assay. When cells are transfected with these expression constructs, the CCHFV proteins are expressed, processed, and assembled into virus-like particles (VLPs) that can bud from the host cell (*Zivcec et al., 2015*). These particles can package a minigenome reporter RNA consisting of viral 5′ and 3′ untranslated regions (UTRs) flanking a reporter gene (NanoLuc) that is transcribed by the viral polymerase upon VLP entry. The UTRs serve as transcription and replication signals specific for CCHFV L-RNA polymerase (*Bergeron et al., 2010*). In addition, NP encapsidation of the CCHFV minigenome RNA is required for its efficient transcription and replication. The GPC is required to generate VLPs morphologically similar to authentic CCHFV (*Zivcec et al., 2015*). VLPs can be transferred to fresh recipient cells, and reporter activity can be read in the recipient cell as a proxy for VLP infectivity. Thus, the VLP assay tests the contribution of NP and L to transcription and replication of viral RNA and the role of GPC in assembly, release, and entry of virus particles. To assess the relative contribution of the individual MT-1303 proteins to VLP activity, we generated VLPs with different combinations of viral proteins from the MT-1303 and prototypic IbAr10200 strains. Surprisingly, VLPs generated with all three MT-1303 proteins failed to generate robust reporter activity when passaged onto naive human cells (*Figure 1B*). VLPs containing MT-1303 NP or L protein individually yielded high reporter activity when complemented with IbAr10200 proteins. These results indicate that the MT-1303 L and NP proteins can recognize their own UTRs and those from the cell culture-adapted IbAr10200 strain to effectively drive transcription. Indeed, MT-1303 L protein was capable of driving transcription from minigenomes derived from other divergent CCHFV strains, though less efficiently than the IbAr10200 L protein (*Figure 1—figure supplement 3*).

Strikingly, MT-1303 GPC failed to produce robust VLP reporter activity with any combination of NP and L protein (*Figure 1B*). In contrast, GPCs from 11 other CCHFV strains representative of all other phylogenetic clades produced high VLP reporter activity (*Zivcec et al., 2017*), and thus attenuation of VLP activity was mainly attributed to the MT-1303 GPC (*Figure 1—figure supplement 4*). Expression of GPC constructs harboring an N-terminal FLAG-tag were comparable between IbAr10200 and MT-1303 strains, indicating that entry deficiency was not due to differences in cellular GPC protein levels (*Figure 1—figure supplement 5A*). We thus concluded that GPC of MT-1303 was expressed normally yet failed to yield robust VLP activity in human cells.

To test which region of the MT-1303 GPC is responsible for poor VLP activity, we generated chimeric constructs of MT-1303 and IbAr10200 GPCs by exchanging the PreGc region (*Figure 2A*). IbAr10200 GPC could not support robust VLP reporter activity when it contained the MT-1303 PreGc region (*Figure 2B*; two-tailed Student's t-test, p=0.0489). Conversely, reporter activity levels from VLPs generated with the MT-1303 GPC harboring the IbAr10200 PreGc region were comparable to those from complete IbAr10200 GPC (*Figure 2B*; two-tailed Student's t-test, p=0.7902). These results indicate that the MT-1303 PreGc region precludes robust VLP activity.

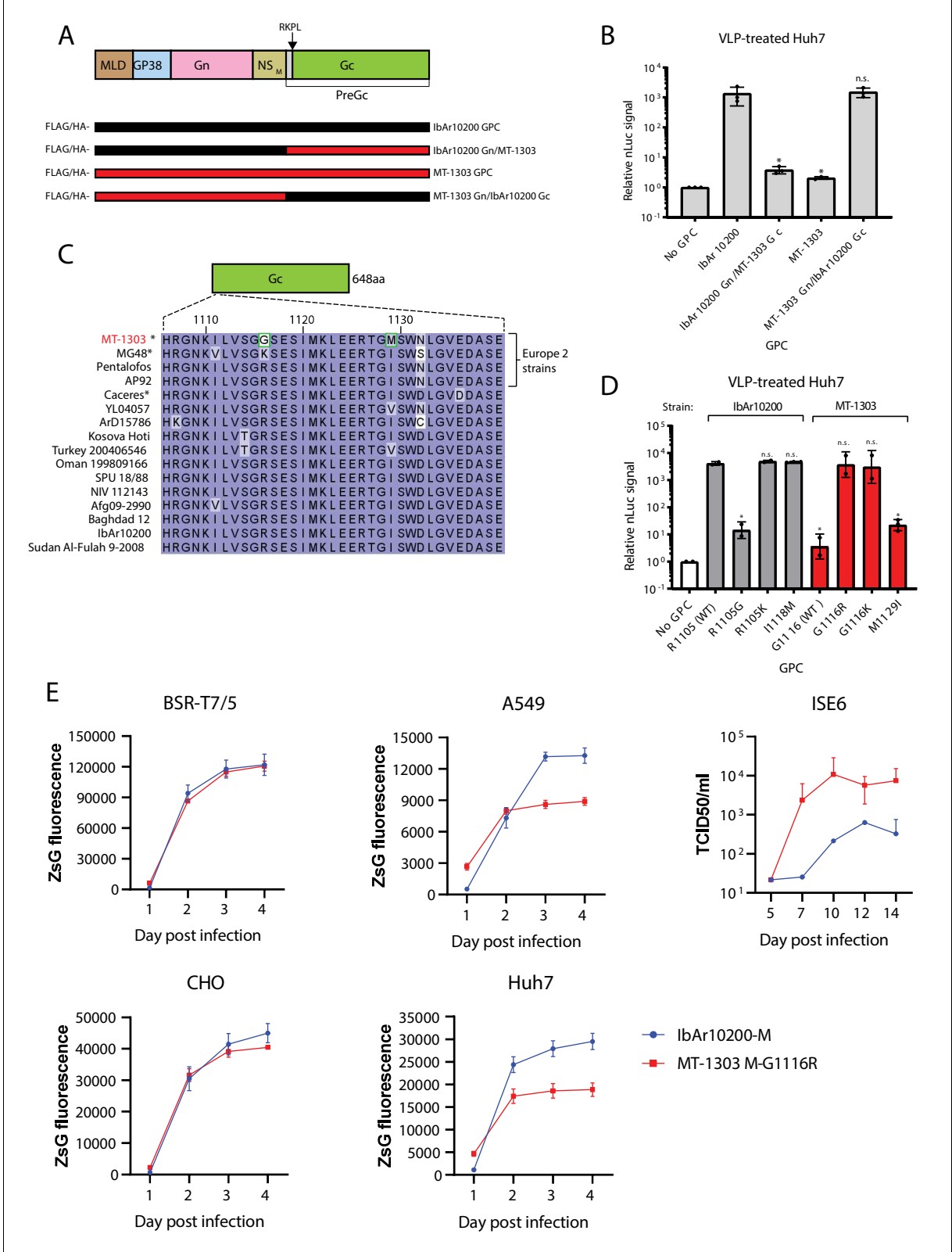

**Figure 2.** Poor infectivity of the MT-1303 strain in human cells is associated with a single amino acid variation in the Gc glycoprotein. (**A**) Schematic representation of the chimeric FLAG/HA-tagged Gc constructs tested. MLD, mucin-like domain. (**B**) nLuc reporter activity levels of VLPs generated using the indicated chimeric Gc constructs in Huh7 cells. *p<0.05 and n.s. = not significant according to the two-tailed Student's t-test; n = 3 independent biological replicates. (**C**) Region of Gc protein sequence alignment from various CCHFV strains highlighting the two amino acids unique

*Figure 2 continued on next page*

*Figure 2 continued*

to the MT-1303 strain. Asterisks (*) denote sequences derived directly from tick sources. Amino acid positions are denoted relative to the entire MT-1303 GPC. (D) nLuc reporter activity levels of VLPs generated using N-terminal FLAG/HA-tagged point mutant GPC constructs in Huh7 cells. *p<0.05 and n.s. = not significant according to the two-tailed Student's t-test; n = 2 independent biological replicates. For all plots, error bars represent standard deviation of the mean. (E) Replication kinetics of recombinant viruses containing IbAr10200 S-ZsGreen and L segments and either IbAr10200 M or MT-1303 M-G1116R segment. BSR-T7/5, CHO, Huh7, A549, or ISE6 cells were infected at an MOI of 0.1, and ZsG fluorescence or TCID50 (ISE6 cells) was measured daily as an indicator of viral replication. Data represents three replicates and error bars represent standard deviation of the mean.

The online version of this article includes the following figure supplement(s) for figure 2:

**Figure supplement 1.** VLPs generated from MT-1303 GPC in Huh7 cells exhibit low reporter activity in tick-derived ISE6-recipient cells.

**Figure supplement 2.** Activity of VLPs containing MT-1303 GPC is not restored at lower temperatures.

## Poor infectivity of MT-1303 in human cells is associated with a single amino acid variation in the Gc region

Next, we investigated whether specific amino acid variations in the MT-1303 PreGc protein sequence correlated with poor VLP reporter activity. We compared the amino acid sequences just following the PreGc cleavage site (RKPL↓) from the MT-1303 strain to other Europe 2 lineage CCHFV strains and 11 other diverse strains that yield robust VLP activity (*Zivcec et al., 2017*; *Supplementary file 2*). We identified two amino acids in the PreGc region, Gly1116 and Met1129, that were unique to MT-1303. The MT-1303 strain contains a glycine (G) residue at position 1116, while all other publicly available sequences of strains obtained from human samples or mammalian cell culture virus isolates have an arginine (R) residue at the corresponding position, indicating a high degree of conservation at this position. MT-1303 also contains a methionine (M) residue at position 1129, whereas other strains possess either an isoleucine (I) or valine (V) (*Figure 2C*). To test the potential functional effects of these unique MT-1303 amino acid variants on viral activity, we generated constructs expressing the MT-1303 GPC harboring single point mutations at positions 1116 or 1129, changing the amino acid at either position to the consensus residue (G1116R and M1129I). Strikingly, G1116R completely restored VLP reporter activity of MT-1303 GPC to levels similar to those of wild-type IbAr10200 GPC (*Figure 2D*; two-tailed Student's t-test, p=0.8397). Consistently, mutating the IbAr10200 GPC residue 1105, which corresponds to MT-1303 residue 1116, from R to G reduced reporter activity to levels comparable to those generated by wild-type MT-1303 GPC. Thus, reduced VLP activity of the MT-1303 GPC can be mainly attributed to a single amino acid variant in the Gc region. The MT-1303 GPC M1129I mutation only modestly increased VLP reporter activity compared to wild-type MT-1303 GPC, and the inverse mutation did not affect IbAr10200 GPC VLP activity (*Figure 2D*).

We next searched all CCHFV PreGc sequences deposited in GenBank to discover any other variants at the position corresponding to MT-1303 GPC 1116. Strikingly, all other sequences had an arginine at this position except for one other strain, MG48, which contained a lysine (K) (*Figure 2C*). The CCHFV-MG48 partial sequence also belongs to the Europe 2 lineage and was derived directly from a tick sample recently collected in the neighboring country of Turkey (*Dinçer et al., 2017*). Importantly, this finding shows that variation at this position is detectable in ticks, whereas CCHFV isolated from mammalian cells or human samples does not contain this variation, possibly because variants with R at this position are preferentially amplified. To test whether substituting K at this position could support VLP activity in human cells, we assessed reporter activity of VLPs generated with the MT-1303 GPC with the G1116K mutation. Like G1116R, the G1116K mutation allowed high levels of reporter activity comparable to those produced by wild-type IbAr10200 GPC (*Figure 2D*; two-tailed Student's t-test, p=0.9028), suggesting that a positively charged residue at this position is important for CCHFV GPC function in human cells.

To confirm these VLP results in the context of replicating virus, we attempted to generate recombinant fluorescent reporter CCHFV using the wild-type MT-1303 M segment or mutant M segment with the consensus R residue using a CCHFV reverse genetics system (*Bergeron et al., 2015*). In this system, cells are transfected with several plasmid constructs that encode the individual CCHFV genome segments under control of the T7 promoter, as well as helper plasmids. To track viral replication, the CCHFV S genome segment is replaced with a variant that also encodes the ZsGreen1

fluorescent reporter (*Welch et al., 2017*). Thus, the CCHFV reverse genetics systems allows functional testing of individual genome segments as well as discrete molecular variants in genome or protein sequence in the context of infectious CCHFV. When complemented with IbAr10200 S and L genome segments, rescue attempts in Huh7 cells were only successful with the MT-1303 M segment encoding GPC-G1116R. Consistent with the VLP results, these data indicate that the MT-1303 M genome segment supports viral replication in human cells when R is present at position 1116, but not when the tick-associated G1116 variant is used. We next attempted to rescue MT-1303 or reassortant CCHFV with IbAr10200 segments. Viruses containing either MT-1303 S, L or mutated M-G1116R could be obtained, emphasizing the importance of R1116 presence in Gc to recover MT-1303 in mammalian cells. MT-1303/IbAr10200 reassortant viruses sequences were confirmed to match the DNA templates used in the virus rescue transfections. The growth kinetics of the recombinant MT-1303 M-G1116R reassortant virus was analyzed in human (Huh7, A549), hamster (BSR-T7/5, CHO), and tick (ISE6) cell lines. Virus generated with MT-1303 M-G1116R or IbAr10200 M displayed similar growth kinetics in hamster cells (*Figure 2E*). In contrast, MT-1303 M-G1116R exhibited reduced viral replication in human cells compared to IbAr10200 M. In tick cells, replicating MT-1303 M-G1116R virus was detected in all tested wells (3/3), while IbAr10200 M virus infected only one out three biological replicates. The single well of tick cells infected with IbAr10200 M virus nevertheless showed delayed growth and reduced viral titers. These results suggest that the M segment of the MT-1303 strain is more successful at infecting tick cells as opposed to IbAr10200 whose replication was more robust in human cells.

Given that the G1116 GPC variant is specific to the MT-1303 sequence obtained directly from a tick source, we next tested whether this variant mediated entry in tick cells. In Huh7 cells, we generated VLPs containing either the IbAr10200, IbAr10200 R1105G (corresponding amino acid to M-G1116 in MT-1303), MT-1303, or MT-1303 G1116R GPC. These VLPs were then transferred to fresh Huh7s or ISE6 cells, and luciferase signal in the recipient cells was measured to assess VLP entry. VLPs containing the MT-1303 GPC did not exhibit higher activity in ISE6 cells compared to Huh7 cells (*Figure 2—figure supplement 1*). One possible explanation for this result is that the G1116 variant renders the GPC temperature-sensitive, functioning more efficiently at lower temperatures required for maintenance in tick cells. To test this possibility, we performed VLP production and VLP incubation with Huh7 or ISE6 recipient cells at lower temperatures to mimic tick cell temperatures. VLPs containing MT-1303 GPC produced and incubated at 28°C did not restore VLP activity relative to the IbAr10200 GPC VLPs (*Figure 2—figure supplement 2*). These results suggest that impaired entry activity of the MT-1303 GPC is not due to temperature sensitivity.

## The MT-1303 G1116 GPC variant impairs membrane fusion

We next sought to identify the replication step(s) associated with poor infectivity of MT-1303 strain with residue G1116 in human cells. CCHFV Gc must be exported out of the endoplasmic reticulum to ultimately mature in the Golgi apparatus, the site of CCHFV assembly (*Zivcec et al., 2016*). First, we examined the subcellular localization of Gc and observed that both IbAr10200 and MT-1303 Gc proteins co-localized with giantin, a marker of the Golgi apparatus (*Figure 1—figure supplement 4*), suggesting proper folding and normal trafficking of MT-1303 Gc to the site of CCHFV assembly. We next investigated whether MT-1303 Gc was being incorporated into membrane enveloped particles. To assess viral particle formation, cells were transfected with constructs expressing C-terminally V5-tagged GPC proteins (*Figure 3A*). The C-terminal tail of the Gc glycoprotein is located within the CCHFV particle (*Figure 3B*). Cell supernatants were ultracentrifuged through a sucrose cushion and the pellets were treated with trypsin to assess the incorporation of V5-tagged Gc into lipid-membrane-protected particles. The V5 epitope was detected in the pelleted fraction from either IbAr10200 or MT-1303 GPC and corresponded to the mature Gc glycoprotein (*Figure 3C*). V5-containing fragments in these pellet fractions were protected from proteolytic treatment, and this protection was lost after solubilization of the VLP envelope with detergent (*Figure 3C*). Together, these data indicate that wild-type MT-1303 GPC is able to promote formation of enveloped VLPs similarly to IbAr10200 GPC.

Since Gc is believed to mediate entry by fusion of the viral envelope with the host membrane, we tested whether MT-1303 Gc could promote membrane fusion. Huh7 cells were transfected with an expression plasmid encoding IbAr10200 or MT-1303 GPC along with a plasmid encoding T7 polymerase. These cells were then incubated at acidic pH before co-culture with cells transfected with a

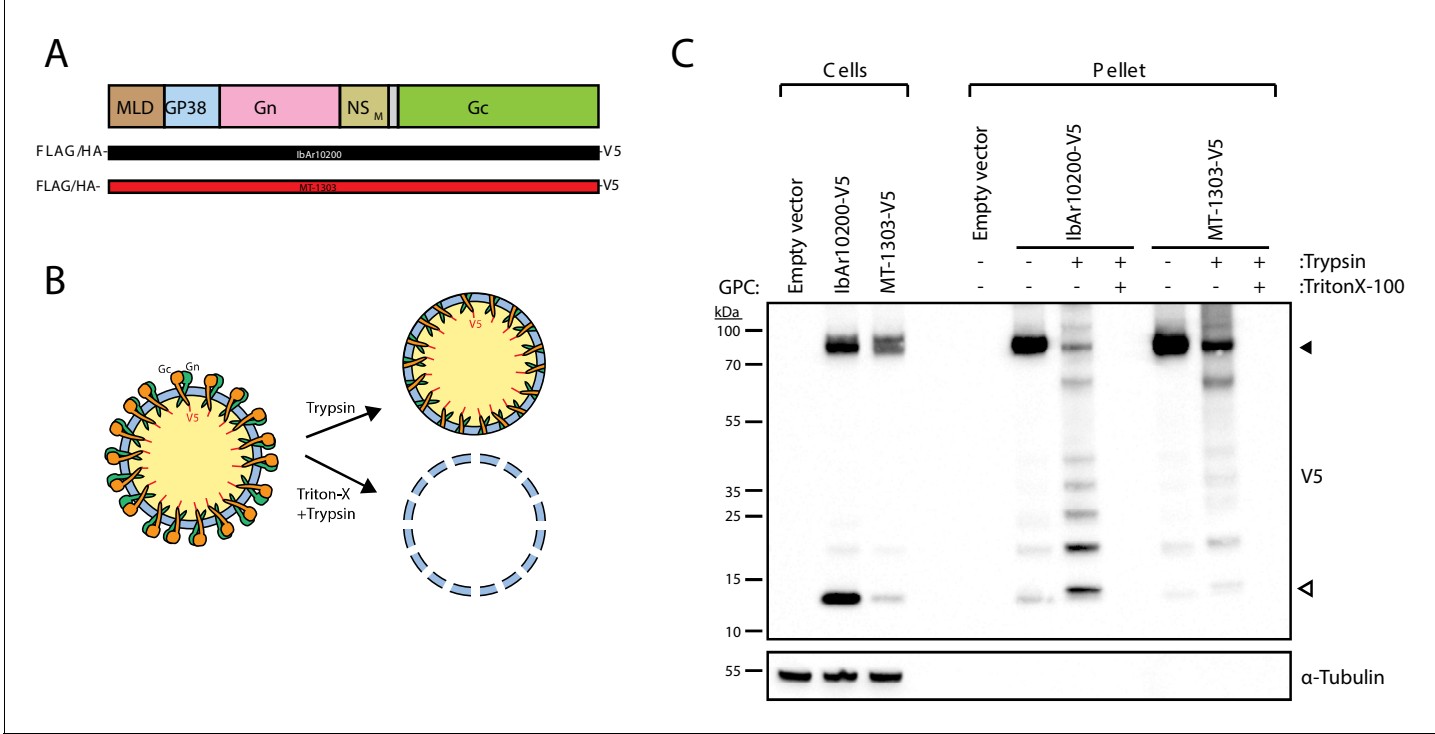

**Figure 3.** The MT-1303 G1116 GPC variant promotes the formation of VLPs. (**A**) Schematic of C-terminal V5-tagged IbAr10200 or MT-1303 strain GPC constructs. (**B**) Schematic representation of how the V5-tagged endodomain of CCHFV Gc glycoprotein is protected from proteolytic degradation. (**C**) Immunoblot analysis of virus-like particles generated by cells transfected with the IbAr10200 or MT-1303 V5-tagged GPC constructs. Closed arrowhead denotes the expected size of the mature CCHFV Gc glycoprotein (75 kDa). Open arrowhead denotes the expected size of the theoretical minimal protected V5 fragment (13.5 kDa). α-tubulin is used as a loading control.

T7 promoter-driven GFP reporter plasmid. In this assay, GPC-mediated cell-cell fusion results in the formation of fluorescent syncytia as transcription of GFP reporter only occurs after cell fusion. Wild-type and N-terminally FLAG/HA-tagged IbAr10200 GPC efficiently promoted the formation of large syncytia when GPC-expressing cells were treated with pH 6 media (*Figure 4A and B*). Closer examination of individual GFP foci revealed that IbAr10200 GPC-induced GFP foci represented large cytoplasmic areas containing multiple nuclei, indicative of membrane fusion of individual cells (*Figure 4A*). Large syncytia were largely absent when MT-1303 GPC was tested compared to IbAr10200 GPC (*Figure 4A and B*; two-tailed Student's t-test, p<0.0001). However, cells expressing the MT-1303 G1116R GPC mutant promoted the formation of large GFP-positive syncytia comparably to IbAr10200 GPC (*Figure 4A and B*), although these syncytia were slightly smaller in area (*Figure 4C*). These data indicate that GPC R1116 is critical for membrane fusion activity, and the G1116 variant severely impairs this activity. Consistent with this, modifying the corresponding position in IbAr10200 GPC to a G residue (R1105G) severely impaired membrane fusion activity (*Figure 4A and B*; two-tailed Student's t-test, p<0.0001).

To test whether a lower pH treatment could rescue fusion activity of the MT-1303 GPC, we performed additional fusion assays at pH 4 and pH 5. Fusion events mediated by the MT-1303 GPC were not restored at these pH levels (*Figure 4—figure supplement 1*). Interestingly, we observed a marked decrease in MT-1303 G1116R GPC-mediated fusion activity at pH 5 compared to pH 6, suggesting that while this mutation can restore fusion activity in the MT-1303 strain, it does not exhibit the same range of pH tolerance as the IbAr10200 GPC. Together, these results indicate that wild-type MT-1303 Gc severely attenuates membrane fusion activity in human cells, leading to poor infectivity. Therefore, we conclude that the tick-associated MT-1303 GPC variant G1116 expressed in human cells poorly supports entry of cells in general.

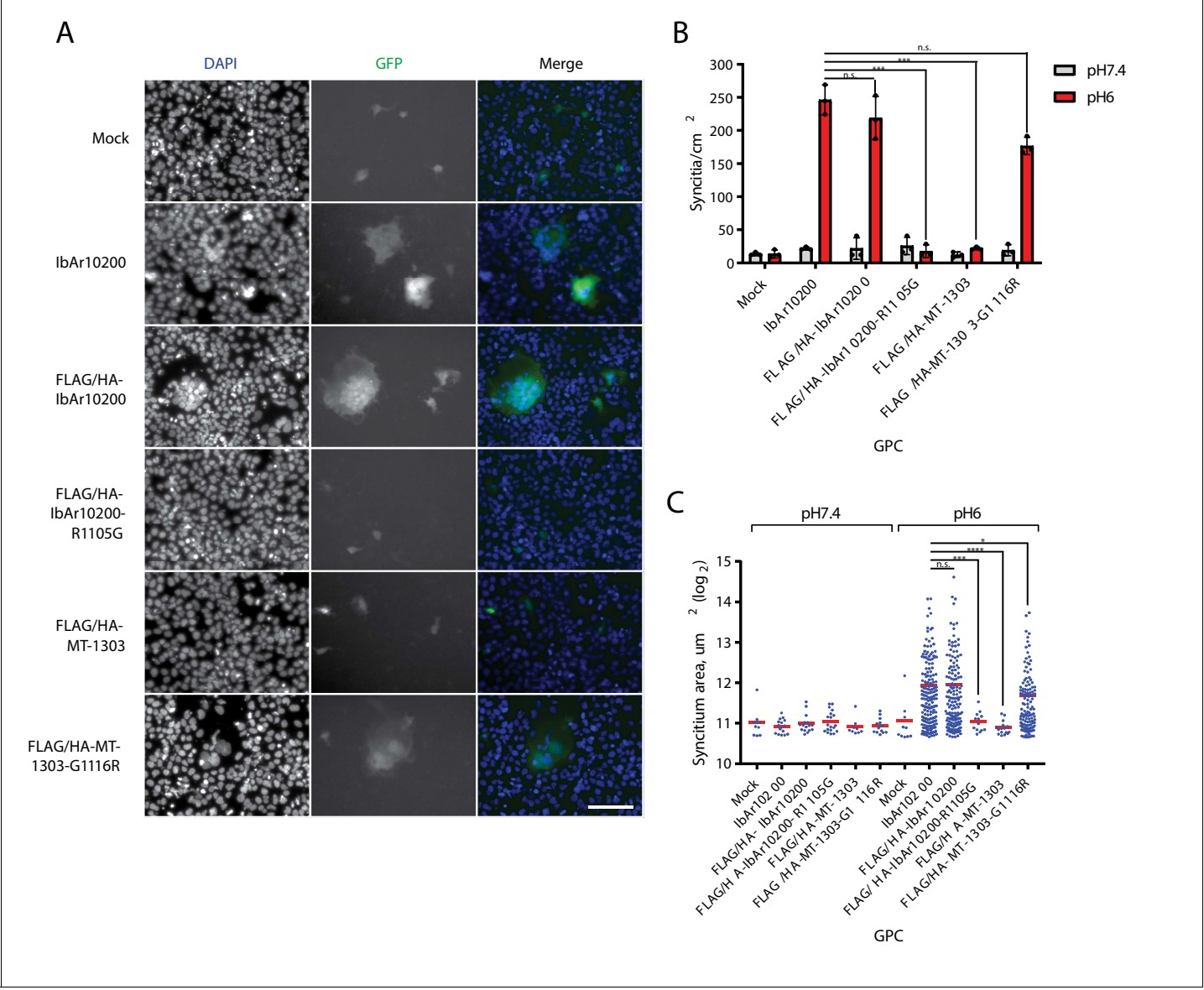

**Figure 4.** MT-1303 G1116 GPC variant impairs membrane fusion. (**A**) Fluorescent images of individual Huh7 syncytia in the GFP cell-cell fusion reporter assay. Cells expressing GPC and T7 polymerase were exposed to pH 6 DMEM media before co-culture with Huh7 cells transfected with a reporter plasmid containing GFP under control of the T7 promoter. Cell nuclei were stained with DAPI. Scale bar represents 100 μm. (**B**) Quantification of syncytia density. Error bars represent standard deviation of the mean of three biological replicate experiments. ****p<0.0001 and n.s. = not significant according to the two-tailed Student's t-test with n = 3. (**C**) Quantification of syncytium area. For each condition, the areas of all of syncytia identified across three biological replicate experiments are shown. *p<0.05, ***p<0.001, ****p<0.0001, and n.s. = not significant according to the Mann-Whitney U-test with n = 3.

The online version of this article includes the following figure supplement(s) for figure 4:

**Figure supplement 1.** MT-1303 GPC-mediated membrane fusion activity is not restored at lower pH.

## Discussion

Most complete genome sequences of CCHFV are derived from human virus isolates typically performed by passaging in mammalian cell lines or in newborn suckling mice. Virus isolation may result in the selection of virus variants that might affect the virus biology. To circumvent potential biases, we obtained three independent CCHFV sequences from ticks collected in South-Eastern Bulgaria. Although this study does not directly address the pathogenicity of CCHFV Malko Tarnovo, it

revealed that certain genetic variants found in ticks have impaired glycoprotein activity in human cells compared to tick cells. Even more dramatic was the discovery that strain MT-1303 GPC fusion activity is severely impaired by the absence of a positively charged amino acid at position 1116.

Unlike mammalian species, CCHFV infection in ticks can be lifelong and results in expansion of the intra-host viral genetic diversity (*Xia et al., 2016*). This could lead to the generation of CCHFV variants like the MT-1303 GPC variant G1116, which would likely be poorly transmitted and amplified by vertebrate host species or would not cause severe disease in humans. Vertebrates can serve as amplification hosts, supporting CCHFV transmission to naive ticks feeding on the same animal. Interestingly, all the CCHFV-infected ticks in this study were collected from the same animal, which might indicate transmission of CCHFV between the ticks during feeding. G1116 may limit CCHFV amplification by vertebrate hosts and consequently reduce transmission to uninfected ticks feeding on the same animal. To gain insights into possible Malko Tarnovo strain transmission between ticks, we analyzed single-nucleotide variations between the consensus genome sequences of the collected CCHFV-positive ticks. Sequences from ticks MT-1302 and MT-1362 were identical, but they differed from the sequence derived from MT-1303 by three nucleotides located in the coding region of the M and L segments; MT-1303 sequences included G1116R in the GPC and A1288T and I1502V variations in the L protein. Reconstruction of the chain of infection is difficult. All three ticks contained similar amounts of CCHFV genome copies ranging from $1.93 \times 10^5 - 3.17 \times 10^6$ copies per mL indicating viral amplification in all ticks. The ticks may either have become infected independently of one another from other sources or from the cow or one of the ticks infected the others during co-feeding on the same host. Since all other ticks sampled in this study were negative for CCHFV, the most parsimonious model would be that the three ticks were infected from the cow or during co-feeding on the cow. Since no blood was collected from the cow upon which the three CCHFV-positive ticks were feeding, whether any of these Malko Tarnovo variants were amplified by the vertebrate host is unknown. However, the strong requirement in mammalian host systems (recombinant CCHFV and VLPs) for a basic amino acid at position 1116 argues in favor of preferential amplification of the M segment from MT-1302 and MT-1362 over that from MT-1303, which has the G1116. Thus, one explanation could be that MT-1303 was infected first and that the R1116 variant was positively selected and enabled replication in the cow. The two ticks MT-1302 and MT-1362 may then have become infected afterwards with the positively selected variant R1116. Of note, the glycoproteins of MT-1302 and MT-1362 were sequenced by next-generation sequencing and position 1116 did not show any nucleotide variations. Unfortunately, MT-1303 could not be re-sequenced by next-generation sequencing due to paucity of sampling material. More studies will be required to fully appreciate the dynamics and importance of CCHFV genetic variations in viral amplification and transmission in vertebrate and tick hosts and the extent to which this influences CCHFV genome evolution and pathogenicity.

Further, characterization of MT-1303 sequence revealed that the wild-type S and L segment sequence could support IbAr10200 strain replication and efficient VLP production, suggesting that MT-1303 M encoding a GPC with G1116 would likely be under more stringent selective constraints to change into an arginine or a lysine as opposed to S and L which supported virus replication in mammalian cells. One limitation of this study resides in the difficulty of addressing MT-1303 GPC function in a tick cell environment due to the technical hurdle of manipulating tick cells to express exogenous proteins. Future studies will require the development of improved methods to transfect and express proteins in tick cells.

In summary, we generated and characterized the full genomic sequences of tick-derived CCHFV strains belonging to the Europe 2 genetic lineage. The M segment was associated with poor CCHFV replication in human cells, contrasting efficient infection of tick cells. The reduced infectivity of the MT-1303 strain in human cells was attributed to reduced Gc-mediated membrane fusion activity. This defect is likely associated with the incompatible cellular environment of human cells as VLPs derived from Huh7 cells were not more efficient at infecting tick cells. Together, our study identifies the presence of CCHFV variants produced by ticks that are likely associated with reduced infectivity in humans. We propose that the detection of CCHFV variants lacking fitness in human cells might contribute to asymptomatic CCHFV infection.

# Materials and methods

## Tick collection and identification

The Strandja Nature Park represents the last remaining temperate forest with evergreen plants in Europe that was not reached by the land-ice during the last ice ages in the Pleistocene and Holocene epochs. It has primeval flora from the Paleogene and Neogene periods, including pontic rhododendron (*Rhododendron ponticum*), oriental beech (*Fagus orientalis*), and various oak species. 1541 ticks were collected by flagging from the vegetation and sampling from livestock, tortoises, and humans in the area of Strandja Nature Park from May to August 2012. Sampling sites were located in the towns of Stoilovo, Silkosiya, Sredoka, Kosti, Bulgari, Sinemorets, Zvezdets, and Malko Tarnovo. Collected ticks were individually cryoconserved. Sex and species were morphologically identified using taxonomic keys (*Babos, 1964*; *Walker, 2014*).

## CCHFV RT-PCR screening

Adult ticks were individually homogenized in 500 µL L-15 medium without additives using six steel beads and a SpeedMill PLUS homogenizer (Analytik Jena AG, Germany). For nymphs and larvae, 200 µL media and 10 ceramic beads were used. Homogenization was performed in two to three cycles for 2 min at a frequency of 30 pulses/sec. The suspension was cleared by centrifugation at 2500 rpm for 10 min at 4°C. Pools were generated by combining 100 µL supernatants of 10 homogenized ticks each according to species, life stage, and sampling site.

Viral RNA was extracted using the QIAamp Viral RNA Mini Kit following the manufacturer's instructions. RNA was transcribed in cDNA using Superscript III reverse transcriptase and random hexamer primers (Invitrogen GmbH, Karlsruhe, Germany). Pools were screened by PCR using primers based on a fragment of the S-segment of CCHFV as described before (*Bergeron et al., 2015*). Subsequently, supernatants of homogenates of individual ticks composing the PCR-positive pools were tested by PCR as described above. PCR products were sequenced by Seqlab (Göttingen, Germany). Sequences were analysed using Geneious Pro v9 (https://www.geneious.com) and compared to other sequences using the NCBI Basic Local Alignment Tool (*Altschul et al., 1990*).

## Full genome sequencing and analyses

For Sanger sequencing, genomes were amplified from tick homogenate using a combination of fragment-specific primers and primers based on the sequence of CCHFV strain AP92 (accession numbers DQ211638, DQ211625, DQ211612). The 3′ and 5′ genomic termini were amplified by rapid amplification of cDNA ends-PCR (RACE-PCR; Roche, Mannheim, Germany). Sequencing was performed by SeqLab. Next-generation sequencing based on long-range PCR products was used for in-depth genomic analyses, using specific primers, the Phusion PCR kit (Thermo-Fisher), and a MiSeq platform (Illumina). Genomes were assembled by reference mapping to the Sanger genomes using Geneious Pro v9. Sequences were analysed in Geneious Pro v9. Signal sequence cleavage sites were identified using SignalP version 4.1 (*Petersen et al., 2011*).

## Phylogenetic analyses

N, GPC, and RdRp genes were aligned using MAFFT, and maximum likelihood (ML) trees were inferred based on 1000 bootstrap iterations in RAxML.

## Virus isolation attempts

All cell cultures were checked for mycoplasma every 20 passages. Freshly seeded $4 \times 10^4$ Vero E6/7 (African green monkey kidney, ATCC, cell identity has been authenticated by STR profiling by ATCC), $4 \times 10^4$ SW13 (Human Adrenal gland cortical small cell carcinoma, gift from DA Bente, University of Texas Medical Branch, originally obtained from ATCC, cell identity has been authenticated by STR profiling by ATCC), $2.5 \times 10^5$ HAE/CTVM8 (*Hyalomma anatolicum* embryo, Tick Cell Biobank, University of Liverpool, cell identity has been authenticated by Tick Cell Biobank [*Bell-Sakyi, 1991*]) and $1.8 \times 10^5$ BDE/CTVM16 (*Rhipicephalus (Boophilus) decoloratus* embryo, Tick Cell Biobank, University of Liverpool, cell identity has been authenticated by Tick Cell Biobank [*Bell-Sakyi, 2004*]) cells in 48-well-plates were inoculated with the supernatant of CCHFV-positive ticks and CCHFV-positive pools filtrated through a 0.45 µm filter under S3 laboratory conditions. Each

cell line was inoculated with 20 µL, 4 µl and 0.4 µL of the respective supernatant and L-15 media without additives was added up to a final volume of 200 µL. After 1 hr of incubation at 37°C and 5% CO2 (VeroE6/7 and SW13 cells) or 28°C (HAE/CTVM8 and BDE/CTVM16 cells) 300 µL of cell line specific medium was added. Cells were observed daily for signs of a cytopathic effect (CPE). Eight days post-infection, 100 µL of supernatants of cell lines SW13 and VeroE6/7 were passaged onto fresh cells. This was repeated five times. The remaining supernatant was centrifuged at 1000 × g for 5 min and stored at −80°C. Infected tick cells were observed for a period of 70 days.

## GenBank accession numbers

The sequences were deposited under the GenBank accession numbers: MK299338 - CCHFV strain Malko Tarnovo-BG2012-T1302 (MT-1032) segment S; MK299339 - Malko Tarnovo-BG2012-T1302 (MT-1302) segment M; MK299340 - Malko Tarnovo-BG2012-T1302 (MT-1302) segment L; MK299341 - Malko Tarnovo-BG2012-T1303 (MT-1303) segment S; MK299342 - Malko Tarnovo-BG2012-T1303 (MT-1303) segment M; MK299343 - Malko Tarnovo-BG2012-T1303 (MT-1303) segment L; MK299344 - Malko Tarnovo-BG2012-T1362 (MT-1362) segment S; MK299345 - Malko Tarnovo-BG2012-T1362 (MT-1362) segment M; MK299346 - Malko Tarnovo-BG2012-T1362 (MT-1362) segment L.

## Cell culture

Huh7 (Apath, LLC, cell identity has been authenticated by Apath, LCC), BSR-T7/5 (gift from KK Conzelmann, Ludwig-Maximilians-Universität, Munich, Germany), and A549 (ATCC, cell identity has been authenticated by STR profiling by ATCC) cells were cultured in Dulbecco's modified eagle media (DMEM) supplemented with 5–10% (v/v) fetal bovine serum, 1% (v/v) non-essential amino acids, 1 mM sodium pyruvate, 2 mM GlutaMAX, and 100 U/mL penicillin/streptomycin at 37°C and 5% $CO_2$. CHO-K1 cells (ATCC, cell identity has been authenticated by STR profiling by ATCC) were cultured in Ham's F12 media supplemented with 10% fetal bovine serum, sodium pyruvate and antibiotics. ISE6 cells (gift from U Munderloh, University of Minnesota) were cultured in L-15B300 supplemented with 10% fetal bovine serum and 5% tryptose phosphate broth at 34°C and 0% $CO_2$ (*Munderloh et al., 1994*). Cell lines were occasionally checked to confirm absence of mycoplasma.

## Minigenome and VLP assays

Minigenome assays were performed in Huh7 cells as previously described (*Zivcec et al., 2015*). Briefly, Huh7 cells were transfected with plasmids expressing CCHFV NP and L proteins and the T7 polymerase along with a minigenome reporter. For the MT-1303 minigenome, a DNA sequence containing the nanoluciferase (nLuc) coding sequence flanked by the UTRs from the MT-1303 L segment was synthesized (IDT DNA) and cloned into the V(0.0) plasmid backbone (*Bergeron et al., 2010*). The NP open reading frame and human codon-optimized L coding sequence of the MT-1303 strain were synthesized (Genscript), amplified by PCR, and inserted into the mammalian expression vector pCAGGS. All transfections were performed with LT1 (Mirus Bio) according to the manufacturer's recommendations. nLuc luminescence was determined in technical triplicate on a Synergy 4 plate reader (BioTek) as a measure of genome amplification 2 days post-transfection; firefly luciferase (pGL3; Promega) was used as a transfection control.

VLPs were generated following the steps above with the addition of a pCAGGS expression plasmid harboring a codon-optimized coding sequence for the GPC during the transfection step. Three days post-transfection, supernatants were harvested and clarified by spinning at 1500 × g for 5 min at room temperature. Clarified supernatants were passaged onto fresh Huh7 or ISE6 cells. nLuc activity was read the following day and normalized to firefly luciferase signal from the transfected cells.

## Recombinant CCHFV generation

Recombinant IbAr10200 CCHFV was generated in the biosafety level four facilities at the Centers for Disease Control and Prevention (Atlanta, GA, USA) as described previously (*Bergeron et al., 2015*). Briefly, Huh7 cells were transfected with plasmids encoding the S, M and L genome segments under control of the T7 promoter, and helper plasmids encoding T7 polymerase, CCHFV NP, and human

codon-optimized L. Supernatants were harvested 4–7 days post-transfection. ZsGreen1-expressing reporter viruses were generated by replacing the S segment with a variant that also contains the ZsGreen1 gene (*Welch et al., 2017*). The MT-1303 M genomic segment was synthesized by GenScript, and the M-G1116R variant was generated using site-directed mutagenesis. Sequences of all recombinant viruses were confirmed by next-generation sequencing (Illumina).

## Generation of chimeric and point mutant GPC expression constructs

Signal peptides of the human codon-optimized pCAGGS-GPC coding sequence of the IbAr10200 or MT-1303 CCHFV strains were replaced with the *Gaussia* luciferase signal peptide (MGVKVLFALICIA VAEAK) followed by the FLAG epitope, a tobacco etch virus cleavage site, and 3 HA epitope repeats. PreGc chimeric constructs were generated by amplifying GPC regions from the parent IbAr10200 and MT-1303 strains by PCR and assembling regions using InFusion cloning. The GPC breakpoint for the PreGc chimeras was after V996 in the IbAr10200 strain (corresponding to V1007 in MT-1303 strain), which lies 41 amino acids upstream of the RPKL site-1 protease recognition sequence. GPC point mutant constructs were generated from parent GPC vectors using site-directed mutagenesis primers coupled with InFusion assembly. Primer sequences are listed in *Supplementary file 4*.

## Immunofluorescence assay

Huh7 cells were transfected with pCAGGS plasmids expressing IbAr10200 or MT-1303 strain GPC and fixed with 1:10 buffered formalin 24 hr post transfection. Cells were permeabilized with $1 \times$ PBS with 0.1% Triton-X100 and blocked with $1 \times$ PBS and 3% BSA. Mouse monoclonal 11E7 was obtained from the Joel M. Dalrymple-Clarence J. Peters USAMRIID Antibody Collection through BEI Resources (National Institute of Allergy and Infectious Diseases, National Institutes of Health), and was used to detect Gc. A rabbit anti-giantin polyclonal antibody (Covance) was used to detect the cis and median Golgi cisternae. Primary antibodies were detected with Alexa Fluor 488- and 594-conjugated secondary antibodies, and cells were imaged on a Cytation5 imager (Biotek) at constant power, integration, and camera gain within each channel.

## GPC expression western blot analysis

Huh7 cell lysates were prepared in Passive Lysis Buffer (Promega) and proteins were separated on 4–12% bis-tris SDS-PAGE gels. Proteins were transferred to nitrocellulose membranes and blocked in PBS-Tween and 5% milk. Mouse monoclonal anti-FLAG M2 (Sigma-Aldrich) and anti-V5 (Thermo-Fisher) antibodies were used to detect N-terminal FLAG-tagged and C-terminal V5-tagged GPC. Anti-mouse HRP-conjugated secondary antibody (Sigma-Aldrich) was used to label primary antibodies, and protein bands were detected using SuperSignal West Dura Fast Western blotting reagent on a Bio-Rad ChemiDoc MP imaging system. An $\alpha$-tubulin (Sigma-Aldrich) antibody was used as a loading control marker.

## Assessment of viral particle formation

Huh7 cells were transfected with expression constructs encoding C-terminal V5-tagged IbAr10200 or MT-1303 GPC proteins or with empty vector control. After 4 days, supernatants were harvested and clarified by centrifugation at $1500 \times$ g for 10 min. Clarified supernatants were layered onto a 20% sucrose cushion and centrifuged at $96,800 \times$ g for 2 hr at 4℃ using the SW-41 Ti rotor (Beckman Coulter). Pellets were dried at room temperature for 10 min and resuspended in $1 \times$ PBS. Aliquots of the pellet sample were either left untreated or were treated with 1 mg/mL trypsin in the presence or absence of 1% Triton X-100 for 1 hr at 37℃. Digestion was quenched by the addition of complete protease inhibitor (Roche) to $1 \times$ final concentration for 2 min at room temperature. Samples were mixed with Laemmli sample buffer with 2-mercaptoethanol (Bio-Rad) to $1 \times$ final concentration, boiled for 5 min at 98℃, and analyzed by Western blotting as described above.

## Cell-cell fusion reporter assay

Huh7 cells were transfected with pCAGGS plasmids expressing IbAr10200 or MT-1303 strain CCHFV GPC or with an empty plasmid control, along with an expression plasmid encoding T7 polymerase. Transfected cells were then incubated for 15 min at 37℃ in DMEM acidified to pH 6, 5, or 4 with 2N

HCl or in neutral pH DMEM (pH 7.4). Huh7 cells transfected with a reporter plasmid encoding GFP under the control of the T7 promoter were dissociated with 1 × PBS and 0.5 mM EDTA and resuspended in DMEM at neutral pH. Resuspended Huh7 cells were added to GPC-expressing Huh7 cells and co-cultured in DMEM at neutral pH for 24 hr. Cells were fixed with 1:10 buffered formalin and counterstained with 1 µg/µL 4',6-diamidino-2-phenylindole (DAPI) to visualize nuclei. Cells were imaged on a Cytation5 imager (Biotek). GFP foci were measured for each pH treatment condition, and the average areas of the foci were determined using ImageJ. Syncytia in experimental wells were called using an area threshold of two standard deviations above the mean area of the foci in the mock-transfected wells at the same pH.

## Statistical analyses

A biological replicate is defined as an additional data point acquired by repeating all steps of an experimental protocol from sample generation to data processing. A technical replicate is defined as an additional data point in which the same sample from an experimental protocol is measured additional times for precision. Means from biological replicate data were compared using unpaired, two-tailed Student's t-tests for normally distributed data sets and Mann-Whitney U-tests for non-normally distributed data sets. p-values<0.05 were considered statistically significant.

## Acknowledgements

We thank Zdravko Dimov and Stoyan Yordanov for support during fieldwork. We are grateful to Lesley Bell-Sakyi and Joy Hecht for advice on culturing tick cell lines. ISE6 cells were graciously provided by U Munderloh. The project was funded by the German Center for Infection Research (TTU 01.801 to CD) and by the Federal Ministry of Education and Research (BMBF) under project number 01KI1716 (to SJ) as part of the Research Network Zoonotic Infectious Diseases. We thank M Peeples for SD1-1 plasmid (pTM1-GFP) and Tatyana Klimova for assistance in editing the manuscript. BLH was supported by an ASM-CDC postdoctoral research fellowship during the course of this work. The findings and conclusions in this report are those of the authors and do not necessarily represent the official position of the Centers for Disease Control and Prevention.

## Additional information

### Funding

| Funder | Grant reference number | Author |
|---|---|---|
| American Society for Microbiology | | Brian L Hua |
| Centers for Disease Control and Prevention | | Stuart T Nichol<br>Christina Spiropoulou<br>Éric Bergeron |
| Federal Ministry of Education and Research | 01KI1716 | Sandra Junglen |
| German Center for Infection Research | TTU 01.801 | Christian Drosten |
| National Institutes of Health | R01AI109008 | Éric Bergeron |

The funders had no role in study design, data collection and interpretation, or the decision to submit the work for publication.

### Author contributions

Brian L Hua, Conceptualization, Data curation, Formal analysis, Funding acquisition, Investigation, Methodology; Florine EM Scholte, Conceptualization, Data curation, Formal analysis, Investigation, Methodology; Valerie Ohlendorf, Marco Marklewitz, Conceptualization, Data curation, Formal analysis; Anne Kopp, Data curation, Formal analysis, Investigation; Christian Drosten, Resources, Funding acquisition; Stuart T Nichol, Resources, Supervision, Funding acquisition; Christina Spiropoulou, Resources, Supervision; Sandra Junglen, Conceptualization, Data curation, Formal analysis,

Supervision, Funding acquisition, Investigation, Methodology, Project administration; Éric Bergeron, Conceptualization, Supervision, Funding acquisition, Methodology, Project administration

## Author ORCIDs
Brian L Hua ⓘ https://orcid.org/0000-0002-7580-3399
Florine EM Scholte ⓘ https://orcid.org/0000-0003-2110-3087
Marco Marklewitz ⓘ http://orcid.org/0000-0003-1828-8770
Christina Spiropoulou ⓘ https://orcid.org/0000-0001-8406-3161
Éric Bergeron ⓘ https://orcid.org/0000-0003-3398-8628

## Decision letter and Author response
Decision letter https://doi.org/10.7554/eLife.50999.sa1
Author response https://doi.org/10.7554/eLife.50999.sa2

# Additional files

## Supplementary files

• Supplementary file 1. Differences between strain Malko Tarnovo from tick T1303 (MTBG2012-T1303) and pathogen strains of other lineages. In parentheses is information about the country and year of strain isolation, the clade into which the strain groups, and the host from which it was isolated. Numbers represent length of the complete nucleotide and amino acid sequences, or lengths of particular domains/proteins, positions of the domains/proteins in the complete amino acid sequence, and pairwise identity of the sequences compared to MTBG2012-T1303.

• Supplementary file 2. Comparison of amino acid substitutions between protein sequences of MTBG2012-T1303 (accession numbers MK299341, MK299342, MK299343) and AP92 (accession numbers DQ211638, DQ211625, DQ211612), as well as between MT-BG2012-T1303 and strains comprising Europe one lineage (Kosovo Hoti, accession numbers DQ133507, EU037902, EU044832; Turkey200310849, accession numbers DQ211649, DQ211636, DQ211623; Turkey-Kelkit06, accession numbers GQ337053, GQ337054, GQ337055; Drosdov, accession numbers DQ211643, DQ211630, DQ211617; Kashmanov, accession numbers DQ211644, DQ211631, DQ211618). Positions of substitutions are indicated by subscripted numbers.

• Supplementary file 3. Cell culture passage history of CCHFV strains pertinent to this study.

• Supplementary file 4. Primers used to generate chimeric and point mutant IbAr10200 and MT-1303 GPC expression constructs.

• Transparent reporting form

## Data availability

All sequencing data have been deposited in GB under accession codes MK299338, MK299339, MK299340, MK299341, MK299342, MK299343, MK299344, MK299345 and MK299346.

The following datasets were generated:

| Author(s) | Year | Dataset title | Dataset URL | Database and Identifier |
|---|---|---|---|---|
| Hua BL, Scholte FE, Ohlendorf V, Kopp A, Marklewitz M, Drosten C, Nichol ST, Spiropoulou C, Junglen S, Bergeron r | 2020 | Crimean-Congo hemorrhagic fever orthonairovirus strain Malko Tarnovo-BG2012-T1302 segment S, complete sequence | https://www.ncbi.nlm.nih.gov/nuccore/mk299338 | NCBI GenBank, MK299338 |
| Hua BL, Scholte FE, Ohlendorf V, Kopp A, Marklewitz M, Drosten C, Nichol ST, Spiropoulou C, Junglen S, Bergeron | 2020 | Crimean-Congo hemorrhagic fever orthonairovirus strain Malko Tarnovo-BG2012-T1302 segment M, complete sequence | https://www.ncbi.nlm.nih.gov/nuccore/mk299339 | NCBI GenBank, MK299339 |

r

| | | | | | |
|---|---|---|---|---|---|
| Hua BL, Scholte FE, Ohlendorf V, Kopp A, Marklewitz M, Drosten C, Nichol ST, Spiropoulou C, Junglen S, Bergeron r | 2020 | Crimean-Congo hemorrhagic fever orthonairovirus strain Malko Tarnovo-BG2012-T1302 segment L, complete sequence | https://www.ncbi.nlm.nih.gov/nuccore/mk299340 | NCBI GenBank, MK299340 |
| Hua BL, Scholte FE, Ohlendorf V, Kopp A, Marklewitz M, Drosten C, Nichol ST, Spiropoulou C, Junglen S, Bergeron r | 2020 | Crimean-Congo hemorrhagic fever orthonairovirus strain Malko Tarnovo-BG2012-T1303 segment S, complete sequence | https://www.ncbi.nlm.nih.gov/nuccore/mk299341 | NCBI GenBank, MK299341 |
| Hua BL, Scholte FE, Ohlendorf V, Kopp A, Marklewitz M, Drosten C, Nichol ST, Spiropoulou C, Junglen S, Bergeron r | 2020 | Crimean-Congo hemorrhagic fever orthonairovirus strain Malko Tarnovo-BG2012-T1303 segment M, complete sequence | https://www.ncbi.nlm.nih.gov/nuccore/mk299342 | NCBI GenBank, MK299342 |
| Hua BL, Scholte FE, Ohlendorf V, Kopp A, Marklewitz M, Drosten C, Nichol ST, Spiropoulou C, Junglen S, Bergeron r | 2020 | Crimean-Congo hemorrhagic fever orthonairovirus strain Malko Tarnovo-BG2012-T1303 segment L, complete sequence | https://www.ncbi.nlm.nih.gov/nuccore/mk299343 | NCBI GenBank, MK299343 |
| Hua BL, Scholte FE, Ohlendorf V, Kopp A, Marklewitz M, Drosten C, Nichol ST, Spiropoulou C, Junglen S, Bergeron r | 2020 | Crimean-Congo hemorrhagic fever orthonairovirus strain Malko Tarnovo-BG2012-T1362 segment S, complete sequence | https://www.ncbi.nlm.nih.gov/nuccore/mk299344 | NCBI GenBank, MK299344 |
| Hua BL, Scholte FE, Ohlendorf V, Kopp A, Marklewitz M, Drosten C, Nichol ST, Spiropoulou C, Junglen S, Bergeron r | 2020 | Crimean-Congo hemorrhagic fever orthonairovirus strain Malko Tarnovo-BG2012-T1362 segment M, complete sequence | https://www.ncbi.nlm.nih.gov/nuccore/mk299345 | NCBI GenBank, MK299345 |
| Hua BL, Scholte FE, Ohlendorf V, Kopp A, Marklewitz M, Drosten C, Nichol ST, Spiropoulou C, Junglen S, Bergeron r | 2020 | Crimean-Congo hemorrhagic fever orthonairovirus strain Malko Tarnovo-BG2012-T1362 segment L, complete sequence | https://www.ncbi.nlm.nih.gov/nuccore/mk299346 | NCBI GenBank, MK299346 |

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
