## [Decision Letter]

**Acceptance summary:**

This work focuses on an important virus, and combines much-needed fieldwork with subsequent follow-up in a high-containment virology facility. A new genetic variant of potential significance to understanding spillover/human infection is identified. We thank you and your team for doing this important work.

**Decision letter after peer review:**

Thank you for submitting your article "A single mutation in Crimean-Congo hemorrhagic fever virus discovered in ticks impairs infectivity in human cells" for consideration by *eLife*. The evaluation of this work has been overseen by Drs. Sara Sawyer (Reviewing Editor) and Neil Ferguson (Senior Editor).

Three peer reviewers and the editors have now discussed the paper. Everyone appreciated the impressive number of ticks (over 1,500) sampled from vegetation and livestock in Bulgaria with the goal to isolate Crimean-Congo Hemorrhagic Fever virus directly from the vector/reservoir. The objective was to learn more about how reservoir isolates differ from human isolates. One important insight did result, which is that the tick-derived virus has a mutation in its GP that is apparently incompatible with the (unknown) human receptor.

Based on our conversations, we are willing to consider a significantly modified version of this manuscript if additional experiments are performed, and if the work is made easier to follow through improvements to the writing and graphics. A significant number of additional experiments is needed in this case, but we would like to leave the door open for this manuscript because the work and topic are so important. If you do decide to address these issues, please aim to submit the revised version within two months or contact *eLife* if an extension will be needed.

Additional experiments required

1) Only three ticks were found to be infected with CCHFV, all from the same animal and all bearing nearly identical virus (the three sequences appear identical from the phylogeny in Figure 1; however in the discussion it is stated that one of the sequences differs at 3 nucleotide positions). One must assume that the cow was infected and that these three ticks had all become infected from the cow, although this is never explicitly addressed (please correct). Given that the other 1,000+ ticks sampled were all negative for CCHFV, this seems the most parsimonious model. It is curious why the authors didn't try to replicate these viruses (or perform the virologic assays described) in cow cells, tick cells, and human cells. This trio would have made the paper more interesting.

2) Relatedly, do G1116 variants retain the ability to fuse tick cells? Do the G1116 VLPs have improved luciferase signal when applied to tick cells? The presented data doesn't exclude the possibility that G1116 could be a lethal mutation, in both ticks and mammalian hosts, and was only detected due to sequencing. Is there any evidence that the arginine or lysine at 1116 is post-translationally modified? Have authors attempted the fusion assay or growth kinetics at temperatures cooler than 37C, e.g. 28C to mimic tick-cell temperatures? Is it possible the G1116 results in a temperature sensitive mutant? Can fusion be restored to the G1116 at lower pH than 6?

3) It is interesting that tick cell lines did not result in isolation of live virus directly from the ticks. Did authors quantify amount of virus in the inoculum they attempted to isolate via qRT-PCR? Did authors quantify any potential virus production during isolation attempts using qRT-PCR? Is it possible that the Malko strain does not cause CPE? Further, would live or hybrid viruses have been recovered if these attempts were made in cow cells? This should be attempted (at least for the hybrid viruses used in 2E if the original tick material no longer exists). If still unsuccessful, how would one interpret this? Surely the GP of this virus can fuse with cow and/or tick cells – if not then the mental model one needs to interpret this paper needs to be explicitly provided.

4) Can a recombinant Malko strain be recovered containing its S and L segment with an M G1116R? The enhanced fitness of the reassortant in ISE6 cells suggests maybe a virus with S, M and L from the Malko strain may have enhanced fitness as well. Further, all such experiments need to be accompanied with additional explanation that make the results significant to non-CCHFV experts.

In the writing and figures, please address the following important points:

A) The reviewers were uncomfortable with link that this variation in Europe 2 strains is the reason why they are less pathogenic. Before making the very exciting statements the authors need to have a decent explanation of why there have been 3 reported fatalities resulting from E2 strains. Similarly, there should be some comment on how, if these E2 tick strains cannot infect humans, it is possible for very high rates of seroconversion (often referred to as sub clinical disease) to be observed? It is expected that the very low level of virus introduced from a tick bit, requires an amplification / infection event in ofder for a detectable antibody level (seroconversion) to be detected – and how would this be possible if the tick viruses cannot infect human cells?

B) Please enhance the Introduction to include more information about CCHFV, its reservoir, its global distribution, and also what is known about its cell biology especially regarding possible receptors.

C) A table is needed to summarize all of the different CCHFV isolated mentioned in the paper. The table should also summarize which were lab-passaged before sequencing (and in what cells), and which were not.

D) Please make sure all isolate names are consistent between the text and the figures, which currently they are not.

E) Figure 1: Where does the tick-isolated virus described in Cajimat et al., 2017, fall on this tree?

F) Figure 2C: please number alignment; keep virus names in alignment consistent with what is in text and in other figures; increase font and reduce darkness of blue color.

G) More interpretation of assays, results, and conclusions is needed throughout. For instance, since few readers of *eLife* are CCHRF experts, the assays used (VLP assay, etc) need to be explained in far more detail, even in the main Results section, so that the reader can follow.

H) Have authors considered attempting isolation in IFNAR mice? Also, please give more information about isolation attempts. (in particular, see requested experiments)

---

## [Author Response]

Additional experiments required1) Only three ticks were found to be infected with CCHFV, all from the same animal and all bearing nearly identical virus (the three sequences appear identical from the phylogeny in Figure 1; however in the discussion it is stated that one of the sequences differs at 3 nucleotide positions). One must assume that the cow was infected and that these three ticks had all become infected from the cow, although this is never explicitly addressed (please correct). Given that the other 1,000+ ticks sampled were all negative for CCHFV, this seems the most parsimonious model. It is curious why the authors didn't try to replicate these viruses (or perform the virologic assays described) in cow cells, tick cells, and human cells. This trio would have made the paper more interesting.

Thank you very much for further thoughts on reconstruction of the chain of infection and further experiments.

The phylogeny in Figure 1 is based on the entire L gene, which is ca. 12 kb in lengths. The genomes of the two ticks MT-1302 and MT-1362 are identical, but they both differ at the same positions from the MT-1303 derived sequence by 1 nucleotide located in the coding regions of the M segment (G1116R variation in the GPC) and by 2 nucleotides located in the L segment (A1288T and V1502I). The scale of Figure 1 is too small to illustrate the latter two nt variations.

Referring to the point from where the ticks have become infected, we included the following sentences in the Discussion:

“Reconstruction of the chain of infection is difficult. All three ticks contained similar amounts of CCHFV genome copies ranging from 1.93x10^5^ – 3.17x10^6^ copies per ml indicating viral amplification in all ticks. The ticks may either have become infected independently of one another from other sources, or from the cow or one of the ticks infected the others during co-feeding on the same host. (…) Of note, the glycoproteins of MT-1302 and MT-1362 were sequenced by HTS and position 1116 did not show any nucleotide variations. Unfortunately, MT-1303 could not be re-sequenced by HTS due to paucity of sampling material. “

For the virus isolation attempts in cell culture we used two different tick cell lines, a monkey (Vero cells) and a human (SW13) derived cell line. However, we did not include cow cells in these virus isolation attempts due to paucity of sampling material. These data are now added to the manuscript. For further information, please refer to point 3.

Although we observed release of the viral RNA from tick cells, this did not yield virus isolates to test replication of Malko Tarnovo strains in cow, tick or human cells. In addition, we performed multiple attempts to rescue CCHFV virus with G1116 GPC variant using our very efficient reverse genetics system. This system requires permissive and highly transfectable cell lines that prohibited its use in tick cells and restricted the study of the G1116 variant produced in mammalian cells. Additional details on our efforts to transiently or stably express CCHFV GPC in tick cells are described in point 2.

2) Relatedly, do G1116 variants retain the ability to fuse tick cells?

We attempted to transfect ISE6 cells derived from the hard-bodied tick Ixodes scapularis to test Malko Tarnovo virus rescue as well as to assess GPC-G1116 function in VLP and fusion activity in ticks. However, transfection efficiency was low and we were met with severe technical challenges in maintaining adherent cells after puromycin selection of transfected cells, thus precluding our ability to perform experiments requiring the efficient introduction of plasmids into the tick cell host. Since, methods to efficiently transfect tick cells are not available further development of tick cell culture genetic systems are required to address the function of this mutation in ticks.

Do the G1116 VLPs have improved luciferase signal when applied to tick cells?

In Huh7 cells, we generated VLPs containing either the IbAr10200, IbAr10200 R1105G, Malko Tarnovo, or Malko Tarnovo G1116R GPC. We passaged these VLPs onto fresh Huh7 or ISE6 cells derived from the hard-bodied tick Ixodes scapularis and measured luciferase signal in the recipient cells. This data is now included in Figure 1—figure supplement 5. Interestingly, VLPs containing the Malko Tarnovo GPC do not exhibit higher activity in ISE6 cells compared to Huh7 cells. One possibility is that the ISE6 CCHFV receptor is functionally distinct from that of Rhipicephalus bursa from which the Malko Tarnovo strain was isolated, and thus the Malko Tarnovo GPC may not mediate efficient entry into ISE6 cells. Another possibility is that the G1116 GPC may mature improperly in human cells perhaps due to the different lipid membrane composition. To address this possibility, we would need to generate VLPs in tick cells. However, due to limitations with ISE6 transfection and maintenance, we were not able to carry out these experiments. These results and the associated discussion have been added to the manuscript text.

The presented data doesn't exclude the possibility that G1116 could be a lethal mutation, in both ticks and mammalian hosts, and was only detected due to sequencing. Is there any evidence that the arginine or lysine at 1116 is post-translationally modified? Have authors attempted the fusion assay or growth kinetics at temperatures cooler than 37C, e.g. 28C to mimic tick-cell temperatures?

Referring to the point that the G1116 variant could be a lethal mutation. We do not believe that the mutation was lethal, as similar CCHFV genome copies/ml of G1116 and R1116 variants were detected in all CCHFV positive ticks (~ 1x10^5^ to 1x10^6^ CCHFV genome copies/ml) indicating that both variants replicated similarly in ticks. In addition, the glycoproteins of both ticks were sequenced by HTS and in both ticks the R1116 position was clean and did not show any showing that exactly this variant was replicating in the ticks. Each nucleotide position was covered by 110 – 150 reads and did not show any nucleotide variants as indicated in Author response image 1:

**Author response image 1. sa2fig1:** 

Regarding the possible modification of R1116, this residue is not exposed to the cytoplasmic environment, where arginine modification can occur. Therefore, we did not attempt to identify as it would be extremely unlikely to modulate Gc ability to fuse with host lipid membranes.

Is it possible the G1116 results in a temperature sensitive mutant?

We performed VLP generation and VLP incubation with recipient cells at lower temperatures that mimic tick cell temperatures. VLPs containing Malko Tarnovo GPC produced at 28C did not restore VLP activity relative to the IbAr10200 GPC VLPs. This data is now presented in Figure 1—figure supplement 6. These results suggest that impaired fusion activity of the Malko Tarnovo GPC is likely not due to temperature sensitivity.

Can fusion be restored to the G1116 at lower pH than 6?

We have performed the fusion assays at two additional pH levels – pH 5 and pH 4. Fusion events mediated by the Malko Tarnovo GPC were not restored at pH levels lower than 6. At pH 4, fluorescent signal was weak across all samples, suggesting that this pH is too low for measuring robust fusion. In all cases, GPC-mediated fusion activity is decreased at pH 5 compared to pH 6. Interestingly, there is a marked decrease in Malko Tarnovo G1116R GPC-mediated fusion activity at pH 5 compared to pH6 indicating that while this mutation can restore fusion activity, it does not exhibit the same range of pH tolerance as the IbAr10200 GPC. This data is now presented in Figure 4—figure supplement 1 and the associated discussion has been added to the manuscript text.

3) It is interesting that tick cell lines did not result in isolation of live virus directly from the ticks. Did authors quantify amount of virus in the inoculum they attempted to isolate via qRT-PCR? Did authors quantify any potential virus production during isolation attempts using qRT-PCR? Is it possible that the Malko strain does not cause CPE? Further, would live or hybrid viruses have been recovered if these attempts were made in cow cells? This should be attempted (at least for the hybrid viruses used in 2E if the original tick material no longer exists). If still unsuccessful, how would one interpret this? Surely the GP of this virus can fuse with cow and/or tick cells – if not then the mental model one needs to interpret this paper needs to be explicitly provided.

Thank you for this comment. Initial virus isolation attempts were made using the two tick cell lines HAE/CTVM8 and BDE/CTVM16 and the two vertebrate cell lines Vero E6/7 and SW13 inoculated with the individual tick homogenates as well as with the pooled homogenates. The amount of virus in the inoculum was not quantified but the amount of virus in cell culture supernatants was measured by qRT-PCR starting from the day of infection over 70 days. No viral genome copies were detectable in vertebrate cells after 7 dpi but in contrast viral genome copies were detectable in tick cells up to day 49 post infection. Fluctuating quantities of CCHFV genome copies were detected in HAE/CTVM8 supernatants inoculated with homogenates of MT-1302 and MT-1362 up to 42 and 49 dpi, respectively, suggesting that these viruses showed low level replication. No variation in genome copies in the sample inoculated with MT-1303 were detected. No morphological changes of tick cells were observed during the entire timeframe. Unfortunately, the apparent replication of the MT strains was only transient which prevented virus isolation.

CCHFV from plasmid DNA can be rescued in few mammalian cell lines such as Huh7 and BSRT7 (Bergeron et al., 2015). The reason for this is unknown but not uncommon as reverse genetic systems for negative strand RNA viruses tend to be technically challenging, this prevented obtaining “hybrid” or wild-type CCHF virus” in cow cells.

4) Can a recombinant Malko strain be recovered containing its S and L segment with an M G1116R? The enhanced fitness of the reassortant in ISE6 cells suggests maybe a virus with S, M and L from the Malko strain may have enhanced fitness as well. Further, all such experiments need to be accompanied with additional explanation that make the results significant to non-CCHFV experts.

In addition to M segment, we generated plasmids encoding the S and L segments of MT-1303. We performed virus rescue to generate S or L reassortment with IbAr10200. Reassortant viruses were obtained in Huh7 cells and passaged on BSRT7 cells. The reassortment and virus sequence were confirmed by NGS and matched the plasmid sequences, confirming that wildtype S and L of MT-1303 supported replication in human and hamster cells. These data are discussed in the revised manuscript.

In the writing and figures, please address the following important points:A) The reviewers were uncomfortable with link that this variation in Europe 2 strains is the reason why they are less pathogenic. Before making the very exciting statements the authors need to have a decent explanation of why there have been 3 reported fatalities resulting from E2 strains. Similarly there should be some comment on how, if these E2 tick strains cannot infect humans, it is possible for very high rates of seroconversion (often referred to as sub clinical disease) to be observed? It is expected that the very low level of virus introduced from a tick bit, requires an amplification / infection event in ofder for a detectable antibody level (seroconversion) to be detected – and how would this be possible if the tick viruses cannot infect human cells?

We agree with the reviewers that more work is required to establish that Europe 2 strains are indeed less pathogenic, more details were added to nuance our thoughts and ensure that the discussion around is centred with the main findings. However, we correctly stated that one only fatality in Iran has been linked to a Europe 2 strain and two mild cases were also reported in Turkey. However, only S sequence from the Iran strain is available and therefore possible reassortment with epidemic strains was not evaluated. Our data suggest that variant with G1116 indeed poorly infects human and even ISE6 tick cells. Technical limitations isolating or rescuing virus or VLPs in ticks’ cells limit the evaluation CCHFV Malko Tarnovo genetic variants produced in ticks or tick cells capacity to infect human cells. These limitations of our study are now highlighted in the text.

Referring to the high seroconversion rates, it has been reported that humans also derive antibodies against insect-specific flaviviruses that are introduced by mosquito bites but cannot replicate in vertebrates. Europe 2 specific immunoassays and increase surveillance are required to more accurately determine if Europe 2 strain are responsible for the seroconversion of humans in countries where antibody prevalence is high and disease burden is low or inexistent.

B) Please enhance the Introduction to include more information about CCHFV, its reservoir, its global distribution, and also what is known about its cell biology especially regarding possible receptors.

Introduction was edited to cover these points.

C) A table is needed to summarize all of the different CCHFV isolated mentioned in the paper. The table should also summarize which were lab-passaged before sequencing (and in what cells), and which were not.

In Supplementary file 3 we now describe the passage history of pertinent strains discussed in the manuscript.

D) Please make sure all isolate names are consistent between the text and the figures, which currently they are not.

Manuscript was revised to ensure the consistent use of isolate names.

E) Figure 1: Where does the tick-isolated virus described in Cajimat et al., 2017, fall on this tree?

We now included the virus previous reference in Cajimat et al (named “Spain_MF547417_2014_tick) in the analysis. It falls within the Africa 3 clade (see Figure 1).

F) Figure 2C: please number alignment; keep virus names in alignment consistent with what is in text and in other figures; increase font and reduce darkness of blue color.

Figure was revised as suggested.

G) More interpretation of assays, results, and conclusions is needed throughout. For instance, since few readers of eLife are CCHRF experts, the assays used (VLP assay, etc) need to be explained in far more detail, even in the main Results section, so that the reader can follow.

We adapted the text to include more experimental details and interpretation of the results.

H) Have authors considered attempting isolation in IFNAR mice? Also, please give more information about isolation attempts. (in particular, see requested experiments)

Detailed information on virus isolation attempts has now been added to the text (please also see points 1 and 3). Virus isolation attempts in IFNAR mice might have been a good option. However, we did not try this, and the material has been depleted for the other experiments.